# ImmunoMet Oncogenesis: A New Concept to Understand the Molecular Drivers of Cancer

**DOI:** 10.3390/jcm14051620

**Published:** 2025-02-27

**Authors:** Reshma Sirajee, Sami El Khatib, Levinus A. Dieleman, Mohamed Salla, Shairaz Baksh

**Affiliations:** 1Faculty of Science, 1-001 CCIS, University of Alberta, Edmonton, AB T6G 2E1, Canada; rsirajee@ualberta.ca; 2Department of Biological & Chemical Sciences, Bekaa Campus, Lebanese International University, West Bekaa, Khiyara 1106, Lebanon; sami.khatib@liu.edu.lb (S.E.K.); mohammad.salla@liu.edu.lb (M.S.); 3Center for Applied Mathematics and Bioinformatics (CAMB), Gulf University for Science and Technology, Kuwait City 32093, Kuwait; 4Department of Medicine, Faculty of Medicine and Dentistry, University of Alberta, 113 Street 87 Avenue, Edmonton, AB T6G 2E1, Canada; l.dieleman@ualberta.ca; 5Department of Pediatrics, Biochemistry and Division of Experimental Oncology, Faculty of Medicine and Dentistry, University of Alberta, 113 Street 87 Avenue, Edmonton, AB T6G 2E1, Canada; 6Women and Children’s Health Research Institute, Edmonton Clinic Health Academy (ECHA), University of Alberta, 4-081 11405 87 Avenue, Edmonton, AB T6G 1C9, Canada; 7BioImmuno Designs, 4747 154 Avenue, Edmonton, AB T5Y 0C2, Canada; 8Bio-Stream Diagnostics, 2011 94 Street, Edmonton, AB T6H 1N1, Canada

**Keywords:** ImmunoMET oncogenesis, inflammation, epigenetics, NFκB, AMPK, leptin, MCT, TAK1, NOD2/RIPK2, RASSF1

## Abstract

The appearance of cancer progresses through a multistep process that includes genetic, epigenetic, mutational, inflammatory and metabolic disturbances to signaling pathways within an organ. The combined influence of these changes will dictate the growth properties of the cells; the direction of further malignancy depends on the severity of these “disturbances”. The molecular mechanisms driving abnormal inflammation and metabolism are beginning to be identified and, in some cases, are quite prominent in pre-condition states of cancer and are significant drivers of the malignant phenotype. As such, utilizing signaling pathways linked to inflammation and metabolism as biomarkers of cancer is an emerging method and includes pathways beyond those well characterized to drive metabolism or inflammation. In this review, we will discuss several emerging elements influencing proliferation, inflammation and metabolism that may play a part as drivers of the cancer phenotype. These include AMPK and leptin (linked to metabolism), NOD2/RIPK2, TAK1 (linked to inflammation), lactate and pyruvate transporters (monocarboxylate transporter [MCT], linked to mitochondrial biogenesis and metabolism) and RASSF1A (linked to proliferation, cell death, cell cycle control, inflammation and epigenetics). We speculate that the aforementioned elements are important drivers of carcinogenesis that should be collectively referenced as being involved in “ImmunoMET Oncogenesis”, a new tripartite description of the role of elements in driving cancer. This term would suggest that for a better understanding of cancer, we need to understand how proliferation, inflammation and metabolic pathways are impacted and how they influence classical drivers of malignant transformation in order to drive ImmunoMET oncogenesis and the malignant state.

## 1. Introduction

### Defining ImmunoMET Oncogenesis

The hallmarks of cancer outline characteristics and proteins important in initiating, driving and maintaining the malignant phenotype [1]. Biological forces of self-renewal, replicative immortality and death evasion are some of the defining features of a cancer cell. Researchers have uncovered several pathways of the hallmarks of cancer, including those involved in immune system avoidance, a critical concept in the carcinogenesis process whereby cancer cells are considered “self”-cells, not foreign cells. Self/non-self-recognition is a clever way to educate the immune system to respond to new encounters. However, acquired mutations in cells over time push cells from being “self-like” to “more foreign-like” resulting in a cancer cell type capable of being recognized by the immune system. There is no way of determining how often this occurs, how robust “foreign-like” it is and thus how the immune system may respond. In spite of these challenges, we have made strides to better understand the reasons for immune system avoidance and how the immune system establishes tumor-immune contacts.

About 30% of cancers originate from a state of chronic inflammation, originally initiated as a healthy immune response that can later trigger altered epigenetics, metabolism and proliferation. The remaining vast majority of cancers have been demonstrated to originate from compounded epigenetic or mutational changes resulting in the observed abnormal growth. In > 70% of all cancers, inflammation is not the initial driver but is a prominent characteristic of the cell that influences homeostatic and growth changes toward the malignant state. For example, how inflammatory bowel disease [IBD]-driven colorectal cancer [CRC] arises versus sporadic colorectal cancer depends on the sequence of molecular events that promote the malignant phenotype. It is widely known that sporadic CRC develops without any defined appearance of inflammatory bowel disease (IBD). As such, it develops with defined stages of most malignancies that include an “adenoma-dysplasia-carcinoma” sequence often in association with epigenetic, genetic and metabolic abnormalities [2,3]. In contrast, IBD-CRC does have a known inflammation link and often drives epigenetic changes and malignant transformations with the sequence of “inflammation-dysplasia-carcinoma”. Thus, IBD-CRC often has higher-grade tumors and sometimes a much poorer prognosis compared to sporadic CRC. Molecular drivers have been uncovered for both, and active research continues to understand the differences. Anti-inflammation therapeutics that truly inhibit the molecular initiator of inflammation and the “cytokine storm” that ensues during IBD will be critical in helping patients with IBD prevent progression to CRC.

In many cases, IBD-CRC occurs at a much earlier age and occasionally presents with fewer polyps and early dysplastic sections in comparison to adenomas and microsatellite instability [4,5]. An unexplored variable often not mentioned in the origin of CRC is inflammation-triggered “metabolic distress” or “metabolic syndrome disorder”, which often occurs in patients with a chronic inflammatory state [6]. We have recently published that the “metabolic syndrome disorder” will need to be resolved as efficiently as resolving the inflammation in order to completely recover from inflammation injury [7]. However, it is uncertain if metabolic syndrome disorder arises or is a consequence of various sporadic cancers.

In cancer, metabolic changes are important in promoting the survival of the cancer cell. Energy and metabolite requirements are limiting factors to cellular growth. It has been known for quite some time that tumor cells switch into aerobic glycolytic fermenters, a phenomenology widely known as the Warburg effect [7]. This ensures a sufficient supply into the pentose phosphate pathway for nucleotide and NADPH synthesis and the required ATP to fuel cancer metabolism. For example, this Warburg effect alters the metabolic requirement of cancer cells to allow malignancy. This is needed due to the hypoxic environment that most cancer cells find themselves in and the need to survive. What is not recognized is when, during the path to malignancy, “metabolic distress syndrome” arises to begin the reprogramming into the Warburg effect. We are now finding that these changes will affect the proliferative and, surprisingly, the inflammatory status of the tumor cell.

Metabolic pathways are influenced by numerous factors ranging from stress, exercise, diet, molecular dyshomeostasis (or signaling pathway disruption), altered epigenetics and, recently, the influence of gut microbiota. A key element regulating metabolism is the AMP-activated protein kinase, AMPK. AMPK, in addition to pyruvate kinase isozyme M2 (PKM2), protein kinase B (also known as Akt), mammalian target of rapamycin (mTOR) and glucose transporters (GLUTs), to mention a few, are key elements in the Warburg effect. Metabolic distress syndrome in diseases, such as IBD, is understudied, but AMPK and mTOR pathway components are important [8]. In addition, the gut microbiota can control fatty acid oxidation in the host via suppression of the AMPKs [9]. Interestingly, a common diabetic drug targeting AMPK, metformin, has been documented to reduce the incidence of CRC in diabetics [10,11]. Thus, we are beginning to realize the link between metabolism, IBD-CRC, the microbiome and inflammation, which may significantly contribute to disease progression towards malignancy.

Genes such as the tumor suppressor, Ras-associated family member 1A (RASSF1A), are frequently epigenetically silenced in human cancers to alter growth pathways. However, they have been demonstrated to have roles beyond tumor suppressor function to include anti-inflammation properties [12]. RASSF1A is an important negative regulator of innate immunity by inhibiting the activation of the NOD2/RIPK2 pathogen recognition pathway unpublished observation). However, it is also an important negative regulator of adaptive immune signaling by restricting death receptor-dependent signaling [13]. Innate immunity is our primary line of defense and responds rapidly to the presence of an invading pathogen or internal injury. It triggers a number of receptor complexes including Toll receptors (TLRs) and NOD2/RIPK2 innate immunity receptors. These activations will ultimately result in the activation of NFκB to drive the upregulation of inflammatory mediators and drive the “cytokine storm”.

Adaptive immunity aids and amplifies the inflammatory state initiated by the immune system. Additionally, adaptive immunity is usually governed by the activation of the resident T cells or recruited T and B cells that can function to promote further NFκB activation and increase the inflammatory response to the affected area [14]. However, in cancer cases, an immune response to attack the cancer cell is not usually mounted due to immune evasion. This immune evasion of tumors is actively being explored and several receptor elements have been found to be “immune checkpoints” that are reviewed elsewhere [15,16,17]. The involvement of immune checkpoint receptors and ligands illustrates the role of the innate and adaptive immune system in modulating the cancer phenotype.

In this review, we highlight some of the emerging molecular players linking metabolism to inflammation and their influence on the proliferation of cancer cells. We suggest that changes in the proliferation, inflammation and metabolism of a cancer cell can be defined as changes in “ImmunoMET Oncogenesis”, a term that is inclusive of three robust drivers of malignant transformation: inflammation, metabolism and proliferation. We speculate that ImmunoMET oncogenesis drives the cancer phenotype and significantly contributes to the difficulties in achieving a high cure rate for cancer patients and possibly others with inflammatory disorders. ***If one is to cure cancer, combinatorial therapy will be needed to resolve inflammation; inhibit epigenetic changes leading to altered proliferation, metabolism and increased inflammation; and reset any abnormal metabolism triggered by metabolic syndrome disorder***. Importantly, therapeutics to resolve the inflammation need to be able to inhibit inflammation close to the apex/initial trigger/driver (s) of inflammation and not to the cytokine pathways that are executors of inflammation to generate the well-known “cytokine storm” [18]. In other words, one must inhibit step 1 or step 2 of inflammation activation in order to fully inhibit the “inflammatory storm” that ensues. Once this can be established, epigenetic changes triggered by abnormal inflammation will be reversed to regain gene expression of tumor suppressor genes, negative regulators of inflammation and possibly positive activators of metabolism.

Although this review will not discuss AMPK, PKM2, Akt, mTOR and GLUTs in great detail, numerous reviews exist to illustrate their importance. They are integral members of bona fide molecular pathways driving proliferation and immune and metabolic dysfunction in cancer and thus serve to be important molecular drivers of ImmunoMET oncogenesis and the cancer phenotype. In this review, we will discuss elements beyond AMPK, PKM2, Akt, mTOR and GLUT that are involved in emerging molecular pathways to drive malignant transformation and the cancer phenotype. We predict that a detailed understanding of the ones we present in this review and future drivers of ImmunoMET oncogenesis will aid in predicting alternate and novel therapeutic schemes and/or in predicting better survival outcomes for patients with cancer or possibly other immune disorders.

## 2. The Role of the Immune System in Cancer: The “Immuno” Component of ImmunoMET Oncogenesis

### 2.1. Pathways of Immunity

The immune system modulates the fight against invading pathogens, restricting autoimmunity or coordinating the repair of damaged tissues. The outcomes of many immune processes are regulated by the combined efforts of both innate and adaptive immunity [19]. The innate immune system is governed mainly by the actions of the pattern recognition receptors (PRRs) involved in the specific recognition of microbial products. Once activated, the PRRs will promote the activation of NFκB, a key transcription factor driving inflammation [20]. Other classes of receptors linked to the innate immune system and NFκB include the classical tumor necrosis factor receptor 1 (TNF-R1) pathway that usually functions independently of the Toll receptor (TLR) pathway (a PRR pathway) to drive NFκB activation. We have published how the tumor suppressor protein, RASSF1A, can interact with the TNF-R1 to drive the activation of this receptor. This usually occurs in the absence of TLR activation and can drive cell death [21]. However, it is known that there is crosstalk between receptor pathways that usually aids in driving inflammation that can lead to multi-tissue inflammation indicative of diseases such as IBD [22,23]. Furthermore, abnormal TNFα signaling pathway is present in several inflammatory diseases such as inflammatory bowel disease Ref. [24], asthma [25] and rheumatoid arthritis [26]. However, anti-TNFα immunotherapy can only treat < 20% of patients with inflammatory imbalances. This suggests that TNF-R1 is only a minor driver of inflammation injury and other inflammation pathways drive the disease state. In addition, other factors such as genetic mutations in the gut, the specific location of inflammation in the gut, tolerization of patients to anti-TNFα therapy and other factors dampen the effectiveness of anti-TNFα therapy over time.

As mentioned earlier, adaptive immunity aids and amplifies the inflammatory state initiated by the immune system and often by the innate immunity system of receptors and signal transducers. Specific recognition molecules drive adaptive immunity, including T and B cells, natural killer cells and macrophages, dendritic and granulocytes, to mention a few. They, like innate immunity cells, have surface recognition molecules to aid in fine-tuning the immune response against specific molecular targets. Typically, recognition by adaptive immune cells will result in tagging the target protein or cell for destruction so the body can rid itself of “non-self” components. Innate immunity usually occurs within seconds of an insult, whereas adaptive immunity is delayed to minutes but will aid in amplifying the response to ensure clearance of “non-self” components.

The activation of NFκB not only promotes inflammation but also modulates the expression of cell death genes, cell growth activators and the expression of proteins involved in metastasis [27]. This centralizes a highly critical role of NFκB in the tumorigenesis process, as the majority of ImmunoMET oncogenesis is also driven by NFκB-dependent inflammation. As such, the activation of immune cells and hence NFκB needs to be turned on and off in a timely manner; thus, NFκB remains a unique therapeutic target for several diseases. That said, NFκB is a heterodimeric protein, and it has been difficult to therapeutically target directly. To circumvent this issue, several “NFκB inhibitors” do exist that target molecular pathways upstream of NFκB responsible for driving its activation [28]. Targeting molecular pathways upstream of NFκB activation has been successful in some cases. To mention a few, upstream activating pathways include TNFα/TNFR1, JAK/STAT, TLRs, MMPs and interleukins (IL-10, IL-23). Moreover, there is a growing list of emerging targets away from these well-known drivers of NFκB-dependent inflammation that this review is focused on. The discussed targets below were selected to illustrate the relevance of inflammatory drivers of NFκB (via the innate or adaptive system) to proliferative and/or metabolic pathways that can significantly contribute to the growth of cancer cells. Although this review will not discuss the arms of the immune system and the role of NFκB in great detail, numerous reviews exist to illustrate their importance.

### 2.2. The Role of the Tumor Microenvironment

The tumor microenvironment is a complex molecular “ecosystem” of immune cells, inflammatory factors, stromal cells, fibroblasts, blood cells and extracellular matrix components, to mention a few [29,30]. These components come together in a temporal manner that varies with the type of cancer and genetic makeup of the individual. This microenvironment is often formed away from the primary site of the tumor as a metastatic entity but can form at the primary site. The tumor microenvironment primarily exists to feed malignant cells and promote their survival. What we know today is that inflammatory mediators (TNFα, IL-6, IL-8 and others) infiltrate the tumor microenvironment in an attempt to provide an attack against abnormal cells. However, what usually arises is genome instability and increased epigenetic activity driven by inflammation [31]. This leads to the inactivation of tumor suppressor proteins, such as RASSF1A and others, to promote increased growth. In addition, infiltrating immune cells such as tumor-associated macrophages (TAMs) and cancer-associated fibroblasts (CAFs) also contribute to the increased malignant growth to fuel the tumor microenvironment [32]. CAFs have key roles in molding the tumor microenvironment and CAF-derived factors are key to this process. CAFs can also affect cancer cell invasion and metastatic potential. TAMs are the prominent innate immune cells in the vicinity of the tumor microenvironment and have supportive roles in driving inflammation in the area. CAFs and TAMs play prominent roles in lung squamous cell carcinoma, head and neck cancer, lung adenocarcinoma and prostate cancer [32].

Once the tumor microenvironment is established, altered metabolism occurs that switches from aerobic respiration to anaerobic respiration (the Warburg effect) to allow the malignant cells in the tumor microenvironment to be an independently growing entity free of dependency on host cells that respire aerobically. Metabolic changes are key to the survival of cancer cells in this environment, and the interplay between inflammation, metabolism and proliferation is reflected in our word “ImmunoMET oncogenesis”. Cancer therapy has focused for years primarily on tumor cells as the targets. However, the focus has shifted to therapeutics to disturb the tumor microenvironment, inhibit the interactions between tumor cells and the stromal components and inhibit the inflammation feeding the tumor microenvironment. Can we explore factors beyond the common regulators of inflammation and metabolism to better control malignant growth? The discussions below will highlight some emerging players in inflammation, metabolism and proliferation whereby therapeutics to those emerging players may interfere with malignant growth and slow down cancer progression. Although this review will not discuss the tumor microenvironment in great detail, numerous reviews exist to illustrate their importance.

### 2.3. Importance of NOD/RIPK2

PRR activation is our first response to the presence of pathogens. The recognition of foreign pathogens mainly relies on 13+ TLR family PRRs that are found on the plasma membrane, microsomes or free-floating in the cytoplasm [33]. The key plasma membrane linked to TLR is TLR4, a receptor utilizing LPS (lipopolysaccharide), a component of the outer membrane of Gram-negative bacteria. Most TLRs activate NFκB via associations with myeloid differentiation primary response 88 (MyD88), TNF-related factor 6 (TRAF6) and interleukin-1 receptor-associated kinase (IRAK) associations. TRAF6/TAK1 associations are then assembled to activate IκBα kinase (IKK) and promote IκBα phosphorylation and degradation with subsequent nuclear translocation of NFκB [34]. The main cytoplasmic PRR, nucleotide-binding oligomerization domain-containing protein (NOD), is stimulated by bacterial products containing muramyl dipeptide (MDP) [35]. Two isoforms of NOD exist, NOD1 and NOD2, that function in a similar manner through the association with the obligate kinase, receptor-interacting protein 2 (RIPK2). This association is known to promote both inflammation and the autophagic response. Following NOD2 activation, RIPK2-associated NOD2 becomes ubiquitinated and phosphorylated (Figure 1) [36]. Once activated it can associate with downstream drivers of NFκB activation, such as TGF-β1 activating kinase (TAK)1, which in turn will activate IκBα kinase (IKK), followed by the phosphorylation and degradation of IκBα in order to drive an NFκB response [37]. In addition, TNF-R1 and TLR4 activation can also lead to TAK1 activation to drive an NFκB, JNK and p38 response [38]. Thus, TAK1 is a critical linker between surface signals and cellular responses in inflammation.

The NOD signaling pathway (especially NOD2-directed biology) is commonly mutated in several inflammatory states, including IBD [39,40]. Most NOD2 SNPs will lead to the loss of function mutations, loss of RIPK2 association and failure to activate NFκB. However, the common CD-linked NOD2 SNP13, 3020insC, is a frameshift mutation leading to truncation of NOD2 and a gain of function mutation that can suppress IL-10 production at the transcriptional level [20]. Surprisingly, neither *Nod2^−/−^* mice nor mice engineered to express the murine homolog of the human *NOD2^3020insC^* variant (that is, *Nod2^2939insC^*) develop spontaneous intestinal inflammation under specific pathogen-free (SPF) conditions. The *Nod2^2939insC^*, but not *Nod2^-/-^*, mice displayed the appearance of experimental colitis using dextran sodium sulfate (DSS), an irritant of the colonic mucosa [41]. These results suggest that NOD2 signaling may be important in initiating the inflammatory response upon DSS-induced injury. Although changes in NOD2 have been associated with Crohn’s disease (CD, a subtype of IBD), recent investigations have linked it with ulcerative colitis (UC) patients as well (UC is another IBD subtype) [42,43]. Therefore, NOD2 can be involved in the appearance of UC and RIPK2 may influence this outcome. Mice with genetic disruption of the *Nod2/Ripk2* have a dysbiotic intestinal flora resulting in altered susceptibility to intestinal inflammation and malignant transformation [43]. The latter suggests a link between NOD2 signaling and host metabolism, affecting the gut microenvironment and altering the chemical composition, which leads to dysbiosis (a predisposition in IBD). In addition, the loss of *Ripk2* has been demonstrated to result in the inability of cells to carry out mitophagy. This results in the enhanced mitochondrial production of superoxide/reactive oxygen species and the accumulation of damaged mitochondria that will trigger caspase-11-dependent inflammasome activation [44,45] that could influence malignant transformation, not only systemically but also in the brain.

Apart from IBD [24], abnormal states and/or mutations in NOD2 have been observed in sarcoidosis [46], pancreatitis [47,48], multiple sclerosis [49] and Parkinson’s disease [50]. Several of these diseases have links to inflammation and/or cancer, thus highlighting the importance of the NOD2/RIPK2 pathway. Further support for the NOD2/RIPK2 pathway was demonstrated recently in a report documenting the presence of a germline mutation in RIPK2 that resulted in a constitutively active form of RIPK2 in osteoarthritis patients [51]. A change in Asn104Asp close to the active site of RIPK2 has been identified in families with osteoarthritis [51,52] and is sufficient to produce a constitutively active kinase that can drive the activation of NFκB in both cell-based assays and in vivo in zebrafish. In a similar manner, an Ile259Thr mutation in RIPK2 was found in the Turkish Japanese population with Behcet’s disease (characterized by blood vessel inflammation throughout the body) that also produced a constitutively active kinase [53,54]. Interestingly, RIPK2 was recently demonstrated to physically associate with α-synuclein clumps in a study, suggesting that “RIPK2 participates in the accumulation of pathologic α -synuclein clumps” to promote inflammation and alter the function of neurotoxic reactive astrocytes [50]. These astrocytes have been linked to numerous neurological diseases, including Alzheimer’s, Parkinson’s, ALS and Huntington’s disease [55,56], to suggest therapeutic benefits from inhibiting RIPK2 [57]. Thus, alterations in RIPK2 can also result in abnormal neuroinflammation and/or systemic inflammation, leading to inflammation-induced malignant changes (especially in the case of IBD/colorectal cancer and pancreatitis/pancreatic cancer). We have observed correlations between RIPK2 activation, AMPK loss and YAP proliferation in inflammatory bowel disease patients [12] to suggest a link between key elements involved in ImmunoMET oncogenesis.

In the context of metabolism, both NOD1 and RIPK2 have been demonstrated to be important in promoting diet-induced inflammation in metabolic tissues (such as in adipose tissues) and for driving insulin resistance and glucose tolerance in a mouse model [58,59,60]. Interestingly, high-fat diet-induced circulating NOD1 activators promote higher fasting glucose concentrations and an increased activation of Akt. Interestingly, these changes were not present when NOD1 was genetically deleted [60]. It was later demonstrated that the pharmacological inhibition with RIPK2 inhibitors regorafenib or gefitinib [61]) is beneficial in treating diet-induced inflammation [59]. These observations were supported by the clinical use of protein tyrosine kinase inhibitors to lower inflammation and blood glucose. It has been further speculated that “dietary saturated fatty acids and bacterial peptidoglycan may synergistically activate NOD1 during HFD by a mechanism requiring further investigation” [60]. Furthermore, the ‘‘infection hypothesis” states that gut microbes or their constituents activate the immune system during chronic inflammatory diseases, including type II diabetes, and have links to NOD2 [60]. A greater understanding of the role(s) for both NOD1/2 and RIPK2 in ImmunoMET oncogenesis is needed to unravel the importance of novel NOD/RIPK2 modulators as alternative avenues for therapeutic intervention.

In a related context, individuals with NOD2 polymorphisms (3020insC and R702W) appear to have an increased susceptibility to CRC [62], lung, laryngeal, urogenital, pancreatic, melanoma, gastric cancers, non-Hodgkin’s and MALT lymphomas [63]. In addition, it has been demonstrated that NOD2 can suppress abnormal growth in breast, kidney, and CRC-based models and animal models [64,65], most likely through its obligate kinases RIPK2. It will be useful to explore the RIPK2 interactome under inflammatory or proliferative states to determine what molecular connections are common under inflammatory or proliferative states. Taken together, the observations reported in Section 2.1 suggest that *NOD2/RIPK2* can impinge on metabolic pathways and can be regulated by tumor suppressor genes, linking inflammation, cancer and metabolism.

***NOD2/RIPK2 and TAK1***. Another upstream regulator of NFκB is the MAP kinase kinase kinase/TGFβ-activated kinase 1 (MAP3K7/TAK1) that is regulated by NOD/RIPK2 and can in turn activate IKK and NFκB and JNK to control cell survival, growth and tumorigenesis (Figure 1) [66]. TAK1 can essentially be activated by TLRs, NOD2, IL-1β, TNFα and TGFβ to drive inflammation. The majority of these activating elements are generated by pathogenic stimuli but can come from surrounding adipose tissue or damaged tissue.

TAK1 has robust links to metabolic control of the main energy sensor, AMPK, through phosphorylation of its threonine-172. Furthermore, TAK1 has also been demonstrated to modulate fatty acid oxidation in the liver via control of AMPK/mTORC1 pathway activation of autophagy, which can also lead to abnormal inflammation and proliferation. Other non-inflammatory roles of TAK1 include involvement in downstream signaling of the adiponectin receptor, leading to fatty acid oxidation and glucose uptake [67], regulation of muscle mass, oxidative metabolism and inverse control of AMPK [68] when compared to other cells such as fibroblasts [69]. Thus, TAK1, in a similar manner to NOD2/RIPK2, can regulate inflammatory and metabolic pathways. Inhibitors of TAK1 have been developed to be selective and potent but have not advanced to clinical trials as of yet [38]. Because of the role of TAK1 in both inflammation and metabolism, TAK1 also has links to cancer [70] and can receive a lot of inflammatory signals generated at the membrane and relay this to downstream regulators of inflammation and proliferation. It can modulate downstream signals linked to MAPK to alter microtubule function and redox homeostasis. Thus, TAK1 activation can occur at all layers of cellular signal transduction, spanning from the membrane proximal layer of receptor–ligand interactions to cytoplasmic signalosome assembly, to result in varied outcomes and phenotypes found in multiple types of cancer. The molecular mechanisms for NOD1 versus NOD2 signaling and NOD2 involvement in autophagy are emerging. But, due to space constraints, these are reviewed elsewhere [71,72].

### 2.4. Importance of Inflammasome

The inflammasome is a multi-protein complex composed of NOD-like domain proteins that can promote the secretion of cytokines such as IL-1β and IL-18 in response to infectious microbes or cellular stress [73]. The nucleotide-binding and oligomerization domain-like receptor, leucine-rich repeat and pyrin domain-containing 3 (NLRP3) inflammasome is the best-characterized inflammasome in humans and mice composed of an adaptor protein and procaspase-1 [74]. The assembly of the inflammasome complex will trigger the cleavage of procaspase-1 to p10 and p20 products that will in turn activate pro-IL-1β and pro-IL-18 into their active products, IL-1β and IL-18 [75]. High amounts of caspase-1 will trigger “pryoptosis” or caspase-1-mediated cell death. It is currently known that, in addition to pathogen stimuli, the inflammasome is also influenced by metabolic changes such as changes in glycolysis, membrane electrolyte fluxes and mitochondrial stress. Because of these aforementioned observations of the inflammasome, a number of recent publications have implicated the activation of the NLRP3 inflammasome to a variety of metabolic diseases including obesity, atherosclerosis and type 2 diabetes [76]. It was speculated that under normal physiology, the NLRP3 inflammasome complex maintained metabolic balance. However, upon chronic activation (e.g., in obesity or hypercholesterolemia), NLRP3 became pathologic and promoted disease [77]. Figure 2 schematically illustrates the complexity of inflammasome biology, including important links to metabolic signaling. It has been documented that K+ efflux agonists enhance mTORC1 activation, increasing expression of hexokinase 1 (HK1) and leading to NLRP3 activation [77]. Pyruvate kinase, isoform M2 (PKM2)-dependent expression of pyruvate dehydrogenase kinase 1 (PDK1) and the resulting phosphorylation of PKR have also been linked to K+ efflux, resulting in the stimulation of inflammasome signaling [73]. The relative expression and activity of PKM2 and PDK1 control the fate of pyruvate and play a major role in the cancer-related “Warburg effect”, mechanistically implying a role of the inflammasome in tumor metabolism.

The results from several groups indicate that inflammasomes can promote and suppress tumor development depending on the types of tumors, the specific inflammasome complexes recruited and the downstream molecular pathways engaged [78,79]. The link to malignancy resides in the role of inflammasomes in controlling inflammation and how chronic inflammation can promote malignant transformation and metastasis. On one hand, it is speculated that inflammasomes will induce a state of immunosuppression and thus fuel abnormal growth in cases of breast and skin cancer. On the other hand, it is a protective factor for inflammation-induced colon cancer by modulating gut barrier immunity [80]. Ultimately, the control of tumorigenesis by inflammasomes may depend on multiple factors such as expression patterns of inflammasome components, effector molecules engaged, tumor location and stages of tumor development [75]. The network of interaction partners illustrated in Figure 2 provides a small snapshot of inflammasome signaling. To better understand the global impact of these signaling pathways, we will need to not ignore how connected cellular signaling pathways are and the links that inflammasome biology may have to metabolism.

### 2.5. Importance of Other Drivers of Inflammation

A discussion of some of the important connections between the immune system and metabolism has been described above. This review strives to give the reader an idea of some unique molecular links between inflammation, metabolism and cancer away from classical pathways. Beyond NOD, TAK1 and the inflammasome, several other regulators of innate immunity have been described that can modulate NFκB and inflammation. These include the TNFα/TNFα receptor [81], TRAFs [82] and the TLR family of innate immunity activators [83]. TLR pathway elements function primarily as our first line of defense against invading pathogens. Most TLRs commonly utilize the adapter protein, MyD88, to communicate external signals to the nucleus. Interestingly, somatic gain-of-function mutations in MyD88 have been found in many hematological malignancies such as activated B cell type diffuse large B cell lymphoma (ABC-DLBCL). This is an aggressive subtype of DLBCL whereby active NFκB is a molecular driver of accelerated growth and remains the least curable form of DLBCL malignancies [84,85]. About 39% of ABC-DLBCL tumors have MyD88 mutations and about one-third have single nucleotide change to result in an L265P amino acid change. This amino acid change is in the Toll/interleukin-1 receptor (*TIR*) homology domain that mediates MyD88/TLR associations. A MyD88 L265P change results in constitutive association with TLRs such that signals are constantly sent to NFκB to promote uncontrolled inflammation and indirect constitutive activation of the JAK/STAT pathway [86]. In the case of ABC-DLBCL, endosomal TLR7/TLR9 are continuously activated in parallel to TLR ligand activation of the B cell receptor and cytokine stimulation of the JAK/STAT pathway, leading to accelerated growth [87]. TLR signaling has varied effects on biology with inflammation as a major effect of TLR activation. However, the strength of the signal of this response can have tumor-promoting effects in the tumor microenvironment and thus can affect the proliferative capacity of that cell. As such, it is a difficult disease to manage and treat.

Moreover, obesity is a well-known metabolic syndrome driven by low-grade inflammation and loss of MyD88. In 2014, Everard et al. [88] provided evidence that MyD88 can function in the intestinal microenvironment to modulate gut microbiota and gut metabolism and boldly showed that the loss of MyD88 in the intestinal cells protected mice from high-fat diet-induced fat mass increase, altered glucose metabolism and elevated inflammation. Furthermore, under high-fat diet-induced obesity in mice, the intestinal-specific deletion of MyD88 resulted in protection against the loss of IL-18 expression in the intestine (a key barrier function element), protection against the loss of expression of the antimicrobial peptide, Reg3g (an element produced by innate immunity cells to regulate host-gut microbiota interactions) and protection against the loss of Foxp3+ CD3 cells in the intestines [88]. Foxp3+ CD3 cells are important in maintaining tolerance to commensal bacteria in the intestines and preventing excessive inflammation in the gut. These observations suggest that maintaining normal levels of Treg cells significantly contributed to the resistance to diet-induced obesity in the intestinal tissue-specific *MyD88* knockout. MyD88 has subsequently been shown to be important in the liver [89] and central nervous system (the hypothalamus) [90]. These studies clearly indicate the importance of MyD88 in inflammation and metabolism that will influence the role of MyD88 in malignant transformation in obese individuals.

## 3. The Role of Metabolism in Cancer: The “Metabolism” Component of ImmunoMET Oncogenesis

Mitochondria control multiple biological processes, including cell death/apoptosis, calcium signaling, oxygen generation and use, energy generation, steroid synthesis, fatty acid oxidation, antioxidant control and hormonal signaling. These important functions maintain our body in a healthy state and protect against injury, external microbial attack and cancer development [91,92,93]. The mitochondrion is a highly relevant organelle for “ImmunoMET oncogenesis” as it provides the energy required for the proper functioning of immune cells, immunological synapse formation and proper and appropriate development of immune tolerance. It also provides key metabolites for AMPK and other players in metabolic control. Therefore, exploring mitochondrial biogenesis should always be considered when exploring the origins and pathology of any disease. One can make the statement that mitochondria have evolved as a central regulator of cellular defense, modulating O_2_ production, apoptosis and the generation of ATP, to mention a few.

Since mitochondria are key organelles involved in energy production, cell signaling and metabolism, they are essential to ATP synthesis, lipid metabolism and nucleic acid metabolism. These pathways and end products generated are utilized and altered by cancer cells as they reprogram mitochondrial energy production to affect tumor development, metastasis and energy sources used to survive [94].

AMPK is a central mediator in metabolism and cellular energy homeostasis both in normal and malignant cells [95]. Many reviews have alluded to the importance of AMPK in cancer. We have discussed the importance of AMPK below and a few less well-known metabolic players linked to cancer below. These later players can significantly affect inflammation, metabolism and proliferation. Although not therapeutic targets, they may be very useful as biomarkers of abnormality associated with the dysregulation of inflammation and growth pathways.

### 3.1. AMP-Activated Protein Kinase (AMPK)

Previous discussions in this section elaborated on the role of the mitochondria in ImmunoMET oncogenesis and several links of AMPK to proliferative (mTOR and Akt) and inflammatory (TAK1) pathways. It is widely known that metabolic pathways are influenced by numerous factors ranging from stress, exercise, diet, genetics and, recently, the gut microbiota. The Warburg effect, the prominent change in the metabolism of a cancer cell, creates a hypoxic environment that most cancer cells find themselves in and need to thrive in [96]. What is now recognized is that during the path to malignant transformation, “metabolic distress syndrome” arises to aid in the reprogramming of the cancer cell. Some key elements influencing metabolic distress syndrome are AMPK, glucose, lactate transporters (GLUTs), MCTs) [96] and mTOR pathway components [97]. In addition, the gut microbiota can control fatty acid oxidation in the host via the suppression of AMPK. Interestingly, a common diabetic drug targeting AMPK, metformin, has been documented to reduce the incidence of colorectal cancer in diabetics, suggesting that resetting metabolic distress syndrome may be a viable therapeutic avenue for treating cancer or inflammatory diseases. We are beginning to see numerous studies confirming the existence of oncoimmuno-metabolic pathways as significant contributors to disease progression toward malignancy.

Indeed, it is now clear that AMPK is a central mediator in metabolism and cellular energy homeostasis both in normal and malignant cells (see Figure 3) [98]. When ATP levels fall, there is a corresponding increase in intracellular AMP levels, and AMPK is activated both allosterically by AMP and phosphorylation of the catalytic subunit (α) by an upstream AMPK kinase. This phosphorylation occurs at Threonine amino acid 172 of the α subunit of AMPK and significantly increases the activity of the kinase. Once activated, AMPK increases energy production by (1) increasing fatty acid availability and uptake [95]; (2) increasing fatty acid oxidation [99]; (3) accelerating glucose uptake [99]; and (4) stimulating glycolysis [99]. At the same time, AMPK switches off energy-consuming pathways such as protein synthesis in an attempt to conserve intracellular ATP. Of importance, AMPK activation is thought to inhibit the growth of cancer cells by switching off protein synthesis and cell proliferation [100]. In addition, AMPK activation has been shown to suppress NFκB, mTOR signaling and p53 activation and induce senescence in human CRC cells. Accordingly, AMPK activators may be useful in the chemoprevention of CRC and other metabolically altered cancers (see Figure 3).

Equally critical to tumor control is the connection that links AMPK to proliferative signaling in light of observations that AMPK can phosphorylate serine-94 [102] and serine-61 [103] of Yes-associated protein (YAP) to inhibit the YAP/TEAD-directed proliferative program and interfere with YAP-directed tumorigenesis. Furthermore, glucose transporter 3 (GLUT3) was identified as a YAP transcriptional-regulated gene involved in glucose metabolism. These results demonstrate that glucose-mediated energy homeostasis is an upstream event involved in the regulation of the Hippo pathway and, potentially, an oncogenic function of YAP in promoting glycolysis. Several studies support a tumor suppressor profile associated with AMPK. High expression levels of AMPK are considered to be prognostic for improved overall survival of CRC patients, and activators of AMPK (such as AICAR and metformin) can inhibit NFκB activation in macrophages [104,105]. Furthermore, AMPK activators can inhibit both acute and chronic experimental colitis (an IBD model) largely by reducing the release of pro-inflammatory cytokines induced by NFκB [106,107,108]. AMPK activation has also been shown to stimulate nuclear factor erythroid-2-related factor-2 (Nrf2) signaling, reduce TNF-α and IL1β levels, lower MMP-9 expression and induce antioxidant enzymes to promote anti-inflammatory actions [109]. As such, indirect pharmacological agonists of AMPK, such as metformin, have been tested for their ability to inhibit cancer cell proliferation, emphasizing the potential importance of AMPK activation in treating cancer in patients. It has also been demonstrated that metformin exerts its anti-proliferative effects on cancer cells via indirect signaling that may be linked to changes in the gut microbiome, intra-organ communication and/or secondary to improving glucose homeostasis. Taken together, metformin use may be of significant clinical importance as good glycemic control and overall metabolic status of the cell may be a requirement for the anti-proliferative action of metformin.

### 3.2. Nitric Oxide and Nitric Oxide Synthase

One of the more well-known gaseous triggers of inflammation is nitric oxide (NO). NO is one of the smallest molecules found in nature and has a free radical gaseous phase. It has been known for many years to be an important signaling molecule to drive molecular changes [110], associate with cytochrome C oxidase (the terminal acceptor in the mitochondrial electron transport chain) and compete with oxygen. These associations result in several biological outcomes, including the generation of reactive oxygen species (ROS), increased mitochondrial membrane potential (Δψ_m_), activation of NFκB (and inflammation) and activation of AMPK to increase glycolytic output (altered metabolism) [111,112]. Dysregulating mitochondrial function results in altered apoptosis, DNA damage control, and pathways linking to metastasis. All of these outcomes will impinge on the ability of cells to maintain homeostasis and affect the appearance of tumorigenesis. It has also been observed that numerous chemotherapeutic drugs will result in increased NO levels, leading to increased inflammation, immunosuppression and a factor for the survival outcome of patients. Interestingly, common anti-inflammatory agents such as curcumin (turmeric) and sulphorafane (found in cruciferous vegetables such as broccoli) can reduce NO levels and aid in counteracting the effects of chemotherapeutic drugs.

It is known that NO is generated from the L-arginine by three functional classes of nitric oxide synthase (NOS): the neuronal NOS (nNOS/NOS1), the endothelial NOS (eNOS) and the inducible NOS (iNOS). It is currently known that NO is produced by both tumor and stromal cells in the tumor microenvironment. It has been demonstrated that iNOS is involved in the activation pathways of matrix metalloproteinase (MMP)-1, -2, -3, -8, -9 and thus can promote extracellular rearrangements related to carcinogenesis and metastasis [111]. iNOS can be transcriptionally regulated by HIF-1α, STAT and NFκB to result in an increased generation of NO in cancer. Thus, iNOS can promote tumor-induced inflammation, leading to increased carcinogenesis [113]. Furthermore, the increased iNOS expression levels in cancers appear to correlate with poor patient survival. iNOS inhibitors inhibit tumor growth, increase sensitivity to chemotherapy and aid in immunotherapy (such as aiding cisplatin efficacy). iNOS expression may be an interesting biomarker for tumor progression and patient prognosis. That said, it is worth mentioning that controversially, iNOS has been demonstrated to have an anti-cancer role and it may well be that the expression levels of iNOS are what dictates its biological function in specific cell types [110]. NO levels can vary considerably with high levels leading to DNA damage, p53 activation and apoptosis to improve chemotherapeutic efficacy and thus anti-cancer effects [114].

On the other hand, eNOS is a calcium-dependent enzyme important for the function of endothelial cells and increased expression of eNOS has been observed in numerous cancers. In addition, eNOS genetic polymorphisms (especially D298E change) have been identified in breast cancer [115], colorectal cancer [116,117] and prostate cancer [118,119] and found to be associated with increased cancer risk. The functional outcome of such changes, although conserved, is yet to be determined. NO has also been demonstrated to be elevated in cervical cancer patients and both eNOS and iNOS have been demonstrated to be involved in the progression of HPV infection to cervical cancer appearance [120]. In addition, it has been demonstrated that Akt/PKB associates and phosphorylates eNOS on S1177, and these associations may involve oncogenic Notch and thus provide a link to growth-promoting pathways [121,122]. In addition, human umbilical vein endothelial (HUVEC) cells can be stimulated with LPS to trigger innate immunity, activation of eNOS (via Akt) and increased expression of iNOS. Thus, both eNOS and iNOS have roles in modulating inflammation and links to proliferation and tumor-regulated pathways (see Figure 4).

### 3.3. Superoxide Dismutase (SOD)

Reactive oxygen species trigger pathways related to apoptosis, DNA damage, growth and inflammation. The first species produced is the superoxide molecule that usually accumulates in the matrix and the intermembrane space of the mitochondria [123]. Superoxide is converted to hydrogen peroxide through the activity of the dismutases SOD1, SOD2 and SOD3. SODs are present in all aerobic living cells, with SOD1 being the major intracellular form of SOD that accounts for the major dismutase activity and ~ 80% of total SOD protein [124].

SODs, especially SOD2, are highly regulated by the deacetylase sirtuin 3 (SIRT3) [125]. SIRT activity is influenced by AMPK and resveratrol (an activator of AMPK and an anti-inflammatory molecule) [126]. Interestingly, it has been reported that SIRT3 levels have abolished or decreased in 87% of breast cancers with concomitant loss of SOD2 expression. SOD1 expression is actually higher in most cancers in order to maintain the integrity of the organelle and modulate the concentration of superoxides [127]. Increasing the flow of H_2_O_2_ originating from the mitochondria will affect AMPK activity and thus metabolically influence aerobic respiration/oxidative phosphorylation and glycolysis. The switch to glycolysis occurs in many cancers and inflammatory diseases linked to cancer [128]. As mentioned previously, this adaptation contributes to the metabolic reprogramming that cancer cells undergo in the tumor microenvironment, which can significantly contribute to the invasive and metastatic nature of cancer cells. SOD2 can contribute to the inflammatory nature of the tumor microenvironment by the activation of NFκB signaling and increasing IKKβ transcription, thus influencing cancer progression, growth and invasion [129]. Very little is known about SOD3, but it is secreted for unknown reasons. It is reduced in several cancers (such as pancreatic cancer), and the overexpression of SOD3 can decrease the growth and invasiveness of cancer cells and reduce the accumulation of hypoxia-inducible factors. Thus, SOD biology represents an interesting link between ROS (in the mitochondria), cellular energetics (AMPK), proliferation (Akt) and inflammation (NFκB activation).

### 3.4. Leptin

Leptin is one of the most important elements in the adipokine network of proteins that influence immune cell biology. It was discovered in 1995 by He et al. [130] and found to be under hypothalamic and systemic (mainly adipose tissue) control mechanisms. The leptin receptor (LEPR) now has a prominent role in both metabolism and inflammation by responding to the secreted levels of leptin and promoting JAK/STAT, NO and matrix metalloproteinase signaling [131]. The influence of leptin on immune biology bridges both innate and adaptive immunity. Leptin activates the expansion of NK, neutrophils, basophils and macrophages to enhance their roles in innate immunity and allergic responses [132]. In addition, leptin drives the expansion of T_H_1, T_H_2 and T_H_17 cells while promoting the inhibition of the growth of T_reg_ cells. T_H_1, T_H_2 and T_H_17 cells have prominent roles in the promotion and maintenance of inflammation and autoimmunity, especially T_H_17 cytokines. The inhibition of T_reg_ cells was mainly via increased mTOR activity that was confirmed in the *Lepr^−/−^* mice [133]. Leptin (encoded by the Ob gene) can also promote the expansion of mature B cells and antibody production in the cell. *Lepr^−/−^* mice have reduced expression of peripheral B cells, and leptin-deficient obese mice (*Ob/Ob*) mice have 70% fewer B cells [134]. Leptin in B cells promoted Bcl2 activation, increased cyclin D1 expression to induce cell cycle entry and elevated the production of pro-inflammatory cytokines. Thus, the ultimate goal of leptin-driven B cell biology is to increase the proliferation and decrease the apoptosis of B cells.

Changes in leptin expression can be regulated by epigenetic mechanisms or promoter-dependent transcription in different cellular settings [135]. Regardless of its genetic regulation, leptin has emerged as a key player in inflammation. Most immune cells will express the leptin receptor and thus are responsive to leptin. However, leptin is mainly secreted by adipose tissue with no production of leptin by epithelial cells. Cancer cells actually express more leptin receptors on their surface than normal cells [131]. Elevated levels of leptin are present in obese individuals, resulting in greater activity of the leptin receptor and increased inflammation and proliferation. Thus, leptin may be an important driver of low-level inflammation states in obese individuals. Abnormal leptin signaling has been documented in several cancers, including those of the breast, endometrium, pancreas, colon, prostate, liver, skin, brain, esophagus, stomach, thyroid gland and ovaries, as well as in leukemia and some bone cancers. Interestingly, leptin can promote increased surface expression of TLR2 to promote inflammation and altered TGF-β1 production, a key element promoting epithelial–mesenchymal transition normally observed during tumorigenesis [136].

Leptin/leptin receptor can promote the downregulation of Bax and the upregulation of Bcl-2 (apoptosis regulators); increased expression of VGEF (angiogenesis); increased activity of Jak/STAT, NOTCH, P-I-3-kinase, cyclin D1 and Rb (proliferation); increased expression of IL-β and TNF-α (inflammation); increased activity of MMP-2, MMP-9 and expression of VCAM-1 (matrix arrangement/metastasis driver); increased activity of HDAC (epigenetic regulator); and increased expression of HIF-1α (hypoxia regulator) [131]. Furthermore, leptin has been documented to increase the expression of breast and pancreatic cancer stem cell markers such as CD44, ALDH1, HER2, Oct-4 and Sox2 [137]. As we continue to learn more about how leptin regulates molecular pathways driving a plethora of biological outcomes, the more we will understand how diet and proper nutrition may play a role in disease appearance (see links in Figure 5). Current findings support the notion that leptin is a master regulator of the immune system and a molecular link between immune tolerance, metabolic function and autoimmunity. The growing list of leptin/leptin receptor targets has shortened the gap between our understanding of inflammation, metabolism and cancer and will offer novel avenues for drug intervention.

### 3.5. Lactate and Monocarboxylate Transporter (MCT)

Monocarboxylate transporters (MCTs) facilitate the passive transport of monocarboxylates such as lactate, pyruvate and ketone bodies together with protons across the plasma membrane [138]. In tumors, MCTs control the exchange of lactate and other monocarboxylates between glycolytic and oxidative cancer cells, between stromal and cancer cells and between glycolytic cells and endothelial cells [138]. Under these exchanges, lactate is a metabolic waste for glycolytic cells, a metabolic fuel for oxidative cells, a signaling agent that promotes angiogenesis and an immunosuppressive metabolite [138]. MCTs are now considered new targets for anti-cancer drugs as they control the trafficking of lactate (both influx and efflux). Lactate is a well-characterized metabolic element, part of the cellular plan to produce energy for normal and neoplastic cells. Other MCTs are receptors for thyroid hormones (MCT8), aromatic amino acids (MCT11) and MCT6, which is a transporter for bumetanide, a sulfamyl category diuretic that is often used to treat heart failure. A comprehensive review of this gene family is found elsewhere [139], and only MCTs 1-4 will be discussed here.

In most normal tissues where lactate is produced, MCT1 is responsible for lactate export across the plasma membrane to the extracellular space. However, in the glycolytic cancer cell and other specific tissues such as white muscle fibers and astrocytes, MCT4 predominates over MCT1 for lactate export [139]. It is speculated that the low affinity of MCT4 for pyruvate could prevent the release of pyruvate, thus allowing for continued conversion of NADH into NAD+ in order to maintain a high glycolytic flux in the cancer cell [140]. The expression of MCTs (especially MCT1) is controlled by NFAT (driven by calcineurin and calcium), AMPK-stimulated PGC1α activity, thyroid hormone T3, p53 and the hypermethylation of CpG islands in the promoter region of the gene [138]. MCT4 expression can also be regulated by HIF-1α and promoter CpG methylation. MCTs are thus highly regulated and have robust roles in both neuronal biology and cancer [138]. Interestingly, haploinsufficient *MCT1* mice demonstrate resistance to diet-induced obesity, insulin resistance and hepatic stenosis. Food intake was suggested to be less important in regulating nutrient consumption. This becomes an important feature of MCT1 activity that illustrates why it is often utilized by cancer cells.

Cancer cells predominantly utilize MCT1 and 4, with a less important role for MCT2 and 3. Several associated chaperones have been discovered for MCT1 and 4, including CD147 and CD44. Some combinations have correlated well with cancer prognosis, such as CD147-MCT1 double-positive bladder tumors having a poor prognosis and unfavorable clinical pathological parameters. Analyses of several cancers have suggested that MCT1 and MCT2 play a role in tumor maintenance, whereas MCT4 would play a role in increasing tumor aggressiveness [138]. Both are thought to be important modulators of metabolism within the tumor microenvironment. MCTs basically allow for glycolysis to operate at high speed by facilitating lactate export. MCT1 is mainly utilized by the oxidative cancer cell while MCT4 is utilized by the glycolytic cancer cell [139]. Using these arguments, lactate is thus a tumor growth-promoting factor and fuels the oxidative metabolism of oxygenated cancer cells. MCTs have also been shown to promote the migration and invasiveness of tumor cells and the stimulation of HIF-α. Interestingly, MCT1 expression was demonstrated to have two Wnt response elements in the upstream region of the promoter to demonstrate Wnt-directed expression [141]. Colon cancer cells were found to be extremely sensitive to genetic manipulation of Wnt-directed MCT1 expression, suggesting an interesting target population that could benefit from efficient MCT1 or Wnt inhibitors [142]. In addition, it has also been determined that glucose deprivation (a situation in cancer cells where glucose is limited) can trigger increased expression of both MCT1 and CD147 [143] and increased MCT1/CD147 complexes to drive oxidative metabolism in the tumor.

Endothelial cells are responsive to lactate levels in many ways similar to those found in malignant cells to promote tumor metastasis but typically only utilize MCT1 [138]. In fact, in endothelial cells, lactate can trigger an NFκB activation, resulting in IL-8 production. Another cell type within the tumor microenvironment is cancer-associated fibroblasts (mainly part of the stromal cells). They utilize MCT4 and glycolysis to generate lactate that will be exported to the tumor cell and fuel tumor growth, inflammation and metastasis. Thus, both endothelial and cancer-associated fibroblasts utilize lactate in glycolytic pathways in the tumor microenvironment. Lactate is also utilized by immune cells and has a peculiar role in immune cells. Activated T cells are glycolytic and utilize pyruvate and generate lactate that needs to be exported. When in communication with another glycolytic cell (such as a cancer cell), lactate exported by the cancer cell will interfere with lactate about to be exported by the immune cell [138]. This interference will result in immune tolerance of the T cell, immunosuppression and a failed T cell response. This is another way cancer cells can evade an attack by the immune system. Furthermore, in myeloid cells, lactate can influence the differentiation of the M2 macrophage cell by activating HIF-α and promoting high VGEF production, angiogenesis and tumor metastasis [144]. Thus, the involvement of MCTs in lactate biology and glycolytic or oxidative metabolism can result in disease states in cells containing expression loss or mutation of MCT1s (see links in Figure 5).

## 4. Discussions

This review has highlighted the importance of understanding the interplay between proliferation, inflammation and metabolism in driving cancer, inflammatory diseases and other diseases. We have introduced a new term “ImmunoMET oncogenesis” to illustrate the importance of relatively less explored elements/links that can influence inflammation, metabolism and/or growth pathways. In our recent findings, we have empirically described a tight link between metabolic dysregulation and excessive inflammation in IBD [12]. In this article, our review proposes that RASSF1A methylation, RIPK2 hyperactivation, sustained YAP activity and AMPK signaling could be potential biomarkers of IBD [12]. In fact, the novelty of the findings presents a starting point to look at IBD from a metabolic and oncogenic perspective. The fact that the AMPK agonist, metformin, rescued DSS insulted mice from inflammation-driven death further potentiates the above link and the term “ImmunoMET oncogenesis”. Metabolic alterations such as changes in glucose metabolism, lipid metabolism and mitochondrial dysfunction definitely contribute to the pathogenesis of IBD and CRC. We are certain that metabolism dysregulation not only affects energy production and cellular homeostasis but also influences the immune response and inflammatory pathways. The maintenance of intestinal mucosal barrier integrity is an energy-demanding process and is highly disrupted in IBD. Furthermore, the bidirectional relationship between metabolism and inflammation should not be underestimated or under-researched; a clear understanding of this interplay will open avenues for new therapeutic interventions to modulate disease severity and improve patient outcomes in IBD and CRC.

### 4.1. Future Directions Linked to ImmunoMET Oncogenesis: The Microbiome

Can the elements discussed in this review influence more than inflammation, metabolism and/or growth pathways (summarized in Figure 5)? An emerging area of research that directly influences ImmunoMET oncogenesis is the microbiome [145,146], a possible hub for ImmunoMET oncogenesis. Over the past two decades, our understanding of how the microbiome can influence disease states has greatly improved. Certainly, a discussion about ImmunoMET oncogenesis is not complete unless there is an inclusion of microbiome analysis. It is known that we have about 10^14^ microbes in our gut and an astounding 1,000,000+ microbial genes to ~ 25,000 human genes in our body or 33+ times more microbial DNA than human DNA! This enormous difference has forced us to understand what constitutes our normal microflora and what microbes are driving disease. Several studies have demonstrated that 90% of our gut microbiota is composed of Firmicutes (49–76%) and Bacteroidetes (16–23%), followed to a much lesser extent by the Proteobacteria and Actinobacteria phyla [147]. Interestingly, as we age, our microbiome content changes from 65% Acinobacteria/Proeobacteria in infancy to 75% Firmicutes/Bacteroidetes at age 80+ [148,149]. Thus, understanding microflora changes will aid in a better understanding of diseases and more effective therapeutic design for diseases such as cancer, obesity, diabetes, IBD, liver disease, Parkinson’s and Alzheimer’s disease [150].

What is challenging today is identifying the few microbial species out of 10^14^ that are driving diseases. More prominent modern-day drivers of disease are *Clostridium difficile* (diarrhea), *Helicobacter pylori* (gastric ulcers) and *Enterobacteriaceae* (liver disease) [151]. Others drive colorectal cancer (*Bacteroides massiliensis*, *Bacteroides ovatus*, *Bacteroides vulgatus* and *Escherichia coli* have also been observed to drive disease from advanced adenoma to carcinoma); esophageal cancer, gastroesophageal reflux disease, Barrett’s esophagus–esophageal adenocarcinoma and GERD (*Helicobacter pylori* is suspected); and atherosclerosis (members of *Chryseomonas*, *Veillonella* and *Streptococcus* are thought to be involved) [152]. These are just some examples of microbial species with links to disease. Understanding how microbes can influence pathways driving the proliferation, inflammation and especially metabolism may be another level of biomarker identification or therapeutic intervention of cancer. Focused reviews can be found elsewhere [153,154,155]. We believe that dysbiosis can be both the cause and the consequence of an altered “ImmunoMET oncogenesis” and thus is positioned as a hub element in the circle of “ImmunoMET oncogenesis”.

### 4.2. Future Directions Linked to ImmunoMET Oncogenesis: 3D Organ Cultures

Innovative models and methodologies have been developed to understand the microbiome and inflammation. Metagenomics has been highly successful in identifying the microbial content differences between normal and disease tissues [156]. Once identified, the challenge would be to empirically test cause and effect. An emerging technology to test the role of single microbes and immune components in a tissue microenvironment is the use of three-dimensional (D) models of tissues [157]. Organoids organize themselves into structures that resemble the 3D environment they originated from and have all the contacts with the surrounding extracellular milieu. For example, 3D intestinal organoids can be formed from human biopsy samples from the lining of the colon of patients by shearing biopsy tissue to release intestinal stem cells (ISCs), mixing the ISCs with matrigel (to provide support for the resultant crypt) and growing in a culture dish. Budding will commence within a few days and a mature colonic crypt (with polarity and an internal lumen structure) will emerge within weeks. These structures will re-create the intestinal crypt composed of stem and differentiated cells. Organoids can grow to >100 M diameter, organize into multiple layers (“crypt structure”) from stem cells at the base of the crypt to differentiated epithelial cells at the top of the crypt and form lumen-like structures. Once formed, organoid structures can be confirmed by immunohistochemical staining for cytoskeletal proteins, extracellular matrix, adherens junction, nuclear stain and proliferation markers. They can be cultured and replated for numerous passages and thus are useful tools for in vivo function and drug discovery.

The combination of 3D organ cultures with high throughput metagenomics to identify relevant microbes will greatly advance our understanding of numerous diseases. Organoids are devoid of microenvironment influences—so what can be learned from this model? Can organoids be utilized to understand the molecular drivers of ImmunoMET oncogenesis? Are these drivers as abnormally regulated in patient organoids as in the organs they arise from? If so, can they promote malignant changes in the absence of microbes or immune cells? Is controlling the intrinsic/mucosal activation of active inflammation or dysregulation of mucosal metabolism sufficient to treat patients? Can organoids be utilized to predict future changes in ImmunoMET oncogenic pathways that could lead to malignant progression, metastasis or disease relapse? These are all intriguing questions that still are currently unanswered.

### 4.3. Future Directions Linked to ImmunoMET Oncogenesis: Emerging Methods of Altering Immune Recognition of Cancer Cells

As mentioned earlier, self/non-self-recognition is vital to the survival of our immune system and can arise from the abnormal behavior of self-cells. Hence, immune recognition will not occur. However, emerging methods have been developed based on our understanding of how self/non-self-cells are identified at the molecular level. The emergence of immune modulation therapy, whereby we trick cells into mounting a response against cancer cells, is one example of manipulating self/non-self recognition. PD1-PDL1, a receptor–ligand system predominantly functioning in the tumor microenvironment to block anti-tumor immune responses/checkpoint responses, was discovered to be involved in self/non-self recognition. It is known that PD-1 is expressed in T cells and PDL-1 is expressed in cancer cells and antigen-presenting cells. Immuno-modulating antibodies to PD1 will sever this association and inhibit the blockade of anti-tumor responses; thus, the immune system will mount a response. This approach, in theory, will work for most cancer types but clinical trials have demonstrated effectiveness of only 20–30%, indicating that we still need to learn more about immune modulation/checkpoint responses [158,159].

Another emerging avenue for treatment is the use of personalized cells called Chimeric antigen receptor (CAR) T cells. CAR-T cells are engineered to have specific proteins on their surface to bind to the patient’s cancer cells. Thus, they result in normal T cells becoming effective killer T cells to kill cancer cells. Car-T immune therapy has been shown to be effective against blood cancers and has been approved by the Food and Drug Administration (FDA) for the treatment of lymphomas, some forms of leukemia and, most recently, multiple myeloma. It might be more effective if CAR-T immunotherapy or immunocheckpoint therapy with anti-PD1 antibodies is administered with therapies to inhibit inflammation and, more importantly, to re-set the altered metabolism that cancer cells usually have.

### 4.4. Future Directions Linked to ImmunoMET Oncogenesis: Understanding Epigenetics

Numerous publications highlight the importance of inflammation and how it can alter molecular pathways and, especially, promote epigenetic changes. Epigenetic changes are usually ignored in publications that propose the use of specific anti-inflammatories such as steroids, anti-IL6, anti-TNFα or IL-1β. However, it should be stressed that inflammation drives epigenetic changes and alters metabolism and proliferation. This has been demonstrated for RASSF1A, one of the most epigenetically silenced genes in human cancers [160,161,162]. In fact, DNA methylation has been well investigated and demonstrated to be a robust hallmark of cancer [163]. Several groups have demonstrated that the most common epigenetically genes silenced in human cancers include RASSF1A (a proapoptotic and tumor suppressor gene), p16^INK4a^ (a cell cycle inhibitor) and death-associated protein kinase (DAPK)1 (proapoptotic gene). The loss of expression of RASSF1A has been suggested to be diagnostic for cancers. However, methylation percentages vary widely within cancer types and in between cancers. This variation can be attributed to the location of methylation probes utilized to determine methylation percentages. RASSF1A has > 30 CpG islands that are methylated and we have demonstrated several hot spots for methylation in various cancers [164]. This would suggest that one has to be aware of these hotspots and focus on the hotspots when developing assays to correlate epigenetic methylation percentages to the loss of RASSF1A and cancer presence. Thus, the correlation with the loss of RASSF1A expression and the loss of p16, IL-8 and TNFα together with biomarkers of inflammation could be an interesting panel to develop for the early detection of cancer.

If you truly need to drive a patient into remission, anti-inflammatories should be selected that can reverse epigenetic changes and promote increased metabolism and decreased proliferation. Although a tall order, evidence does suggest these inhibitors may exist to give the aforementioned desired outcomes. This review may stimulate discussions about these desired outcomes and options available to promote better, more targeted treatments for patients. The complexity of cancer and the numerous regulatory factors for inflammation, metabolism and proliferation will need both 3D and 2D approaches and the development of smart and effective combination therapies to treat cancers. Figure 5 attempts to link the emerging and established elements together.

## 5. Conclusions

This review has highlighted how inflammation, metabolism and proliferation are all connected, sometimes sharing common signaling partners (Figure 5). This information is vital in designing next-generation ImmunoMET oncogenic arrays in order to obtain a well-rounded snapshot of abnormality in the disease state. Biomarker identification may arise from these analyses and aid in preventing disease relapse, preventing progression to malignancy and increasing the overall survival of patients with cancer, inflammatory diseases and other diseases.

## Figures and Tables

**Figure 1 jcm-14-01620-f001:**
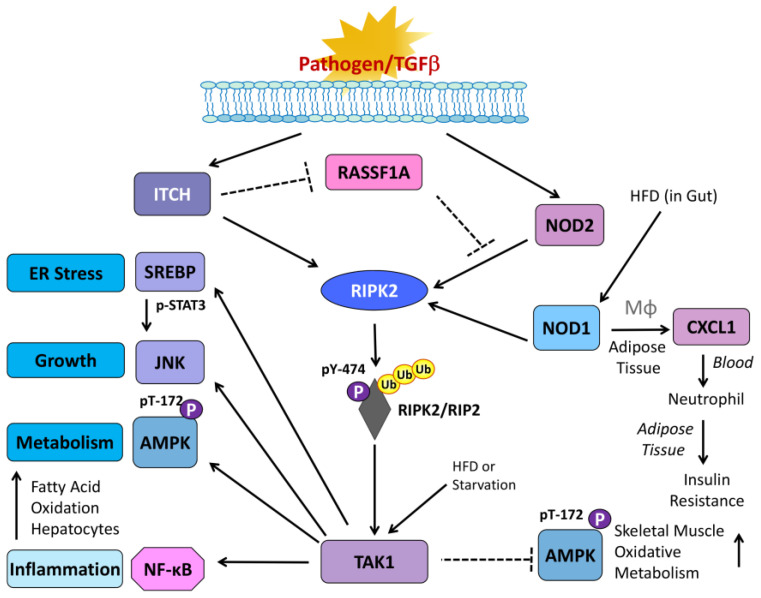
Model for RIPK2 signaling. RIPK2 is emerging as a key player in driving inflammation, metastasis and possibly metabolism. It has been documented that RIPK2 kinase activity is controlled by phosphorylation and K-63 ubiquitination events that are controlled by associations with NOD, mainly NOD2, in response to pathogen exposure. We have unpublished evidence that the tumor suppressor, RASSF1A, can restrict access of RIPK2 to NOD2 and thus regulate RIPK2 kinase activity (unpublished information). In addition, the E3 ligase, ITCH can ubiquitinate RIPK2 upon TGFβ addition to stabilize RIPK2 and drive inflammation and ubiquitinate RASSF1A to degrade it (and thus promote inflammation). Once activated, TAK1 is the immediate downstream target of active RIPK2 that can then in turn promote the activation of NFκB and modulation of AMPK. Active TAK1 and AMPK will lead to modulation of numerous other signaling events as illustrated, with AMPK described as a key metabolic sensor. NOD1 activation with RIPK2 can lead to CXCL1 secretion from macrophages (MΦ) to result in insulin resistance in specific cell types. High-fat diets or starvation conditions will lead to the activation of NOD1 and TAK1, thus providing another link between inflammation and metabolic perturbations. Please note that this schematic only contains pathways relevant to the discussion in this review. Arrows represent activating signals while others represent inhibitory outcomes. HFD, high-fat diet; ER, endoplasmic reticulum.

**Figure 2 jcm-14-01620-f002:**
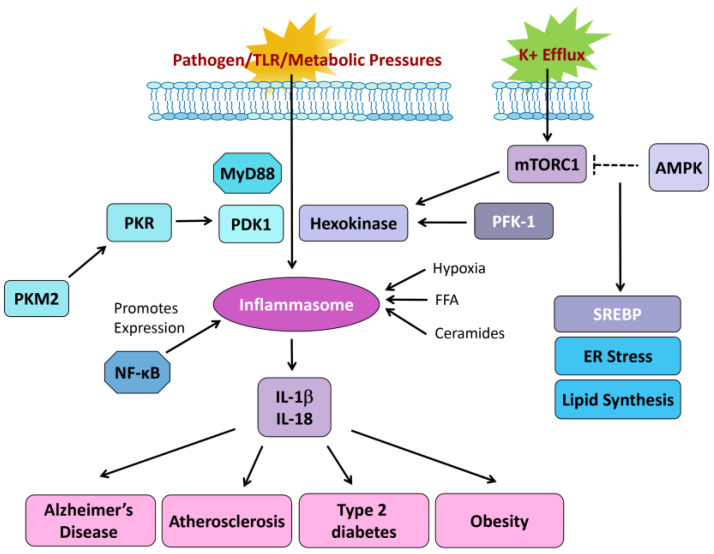
Inflammasome signaling. Metabolic activation of hexokinase, PDK1 and membrane-bound MyD88 (in response to pathogen attack, TLR activation or metabolic stressors) can promote classic inflammasome formation to produce both IL-1β and IL-18 to influence inflammatory biology (as reviewed elsewhere [73]) or pathways leading to unexpected diseases as illustrated and described in this review. Arrows represent activating signals while others represent inhibitory outcomes. mTORC1, mammalian target of rapamycin complex 1; PDK1, pyruvate dehydrogenase kinase 1; PFK-1, phosphofructokinase-1; PKM2, pyruvate kinase, isoform M2; PKR, dsRNA-activated protein kinase; FFA, free fatty acid. Adapted from Prochnicki and Latz (2017) [73] and from Haneklaus and O’Neill (2015) [77].

**Figure 3 jcm-14-01620-f003:**
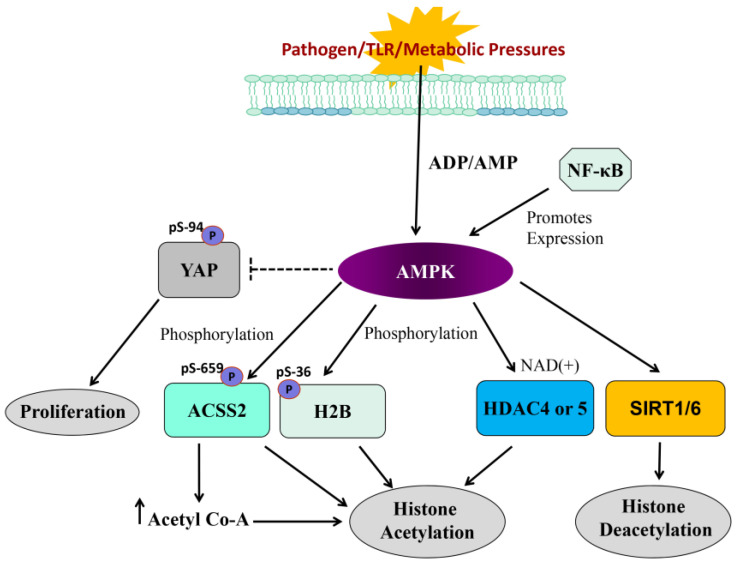
AMPK-related pathways. AMPK is a master regulator of metabolic pathways that can influence inflammatory and growth pathways as reviewed elsewhere [95,101]. AMPK activation can be triggered by pathogen attack, TLR activation or NFκB activation. AMPK effect on histone modifications (and with effect on sirtuins) links a metabolic sensor to a modifier of DNA and gene transcription. The effect on histones occurs in parallel with the AMPK control of serine-94 on YAP and possibly S61 and threonine [T] 119 phosphorylation on YAP. These YAP phosphorylation events will interfere with its ability to be a transcriptional co-activator and mostly be retained in the cytoplasm. Glucose Transporter Type 3 (GLUT3) expression is a target of YAP biology and will be affected by varying AMPK activities, altering glucose access to cancer cells. Thus, these observations illustrate the importance of AMPK beyond metabolic control. ASCC2, Activating Signal Cointegrator 1 Complex Subunit 2; HDAC, histone deacetylase; SIRT, sirtuin. Arrows represent activating signals while others represent inhibitory outcomes.

**Figure 4 jcm-14-01620-f004:**
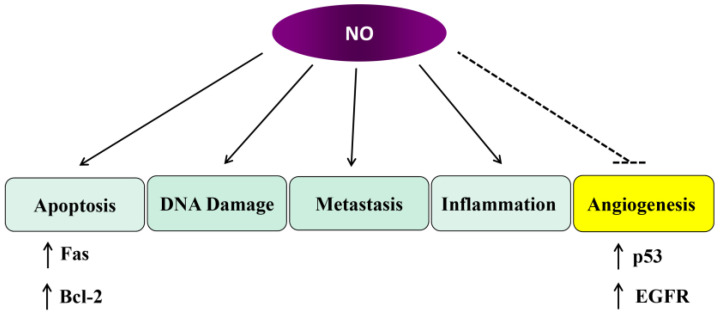
Non-inflammatory effects of NO. NO has always been documented to be a robust activator of inflammation. Investigation over the past decade or so has uncovered several novel NO signaling pathways linking to numerous biological pathways, some activating and some inhibitory, as illustrated. Arrows represent activating signals while others represent inhibitory outcomes.

**Figure 5 jcm-14-01620-f005:**
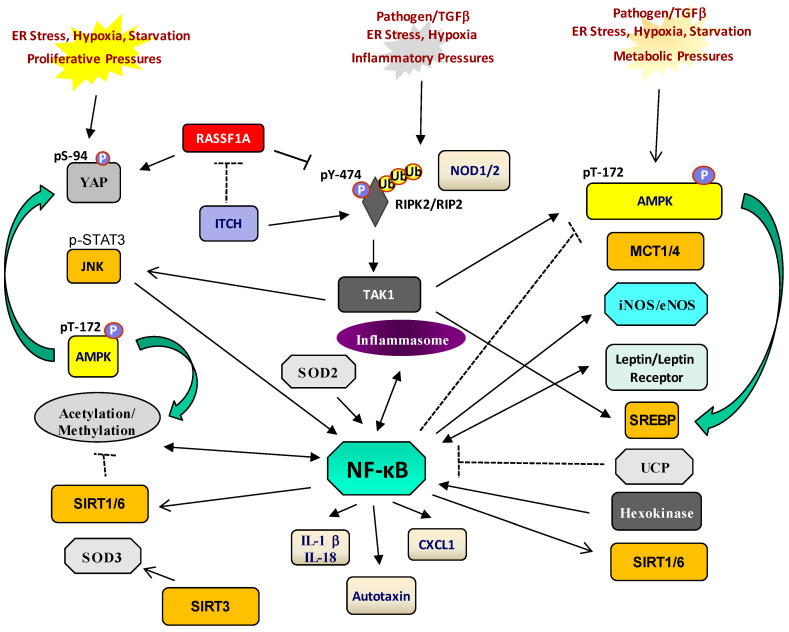
ImmunoMET oncogenesis model of molecular concepts documented in this review that are driving proliferation, inflammation and metabolism in cancer and possibly inflammatory diseases. Arrows represent activating signals while others represent inhibitory outcomes. Please note that factors driving these three pathways are not limited to the indicated elements but only reflect what has been discussed in this review.

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
