# Peer review of "ImmunoMet Oncogenesis: A New Concept to Understand the Molecular Drivers of Cancer"

_jcm, 2025, doi:10.3390/jcm14051620_

Round 1

Reviewer 1 Report

Comments and Suggestions for Authors

This manuscript presents a thorough exploration of the interplay between inflammation, metabolism, and proliferation, introducing the concept of "ImmunoMET oncogenesis" to unify these pathways in the context of cancer and other diseases. While the manuscript is well-researched and offers valuable insights, there are areas where additional clarity, detail, and examples could enhance the overall impact. Below, I provide specific suggestions for improvement:

Line 67: Provide more detail about how IBD-CRC differs molecularly from sporadic CRC to enhance understanding.

Line 72: Reword "if the metabolic syndrome disorder presents or is a consequence" for better understanding.

Line 89: The statement on gut microbiota's role in fatty-acid oxidation could use a concrete example or citation to make it more impactful.

Line 107: Consider referencing a specific example of immune checkpoint therapies to link to current clinical applications.

Line 127: Avoid redundancy by summarizing AMPK, PKM2, Akt, mTOR, and GLUT in one concise statement instead of listing them again.

Line 138-139: Clarify how TNF-R1 functions “independently” of the TLR pathway. Provide a brief example if possible.

Line 142: Expand on why anti-TNFα immunotherapy is only effective for <20% of patients. Mention potential resistance mechanisms or alternative pathways involved in inflammatory imbalance.

Line 171: Expand on "inflammatory mediators (TNFα, IL-6, IL-8)" by briefly describing their specific roles in the tumor microenvironment.

Line 175: Explain how tumor-associated macrophages (TAMs) and cancer-associated fibroblasts (CAFs) promote tumor growth with a specific example.

Line 282-283: Explain how TAK1 links to both TLRs and NOD2 to drive inflammatory signals.

Line 344: Briefly explain how TNFα, TRAFs, and TLR pathways differ in their contributions to NFκB modulation to provide context.

Line 355: Explain why TLR7/TLR9 activation contributes to treatment challenges in ABC-DLBCL, with a focus on therapeutic implications.

Line 364: Explain why Foxp3+ Treg cells are significant in preventing inflammation and obesity.

Line 383: Replace "hijack the energy generation machinery of mitochondria" with "reprogram mitochondrial energy production" to maintain a professional tone.

Line 471: Clarify the contradictory roles of iNOS by providing an example of its anti-cancer effects and discuss potential factors that dictate its dual functions.

Line 514: Briefly explain the JAK/STAT pathway and its role in inflammation and cancer to provide context.

Line 532: Discuss leptin’s potential role as a biomarker for low-grade inflammation in obesity and cancer.

Author Response

Response to Reviewer #1

We thank the reviewer for their comments and suggestions. We have made the required corrections as indicated below.

Line 67: Provide more detail about how IBD-CRC differs molecularly from sporadic CRC to enhance understanding.

SB: We have added a few sentences about the differences between these two and references. Detailed differences is beyond the scope of this review.

Line 72: Reword "if the metabolic syndrome disorder presents or is a consequence" for better understanding.

SB: Thank you for this correction. We have changed to “However, it is uncertain if metabolic syndrome disorder arises or is a consequence of various sporadic cancers.”

Line 89: The statement on gut microbiota's role in fatty-acid oxidation could use a concrete example or citation to make it more impactful.

SB: We have added a citation here.

Line 107: Consider referencing a specific example of immune checkpoint therapies to link to current clinical applications.

SB: We have cited this already with the statement, “This immune evasion of tumors is actively being explored and several receptor elements have been found to be “immune checkpoints” that is reviewed elsewhere [12-14].”

Line 127: Avoid redundancy by summarizing AMPK, PKM2, Akt, mTOR, and GLUT in one concise statement instead of listing them again.

SB: I have looked over my discussions with these markers and AMPK is detailed and the others are stated by not overstated. PKM2 appears in figure 2, Akt appears above figure 4, and the others just once also. Can the reviewer clarify what he/she means?

Line 138-139: Clarify how TNF-R1 functions “independently” of the TLR pathway. Provide a brief example if possible.

SB: I have added a few sentences and references to address this point.

Line 142: Expand on why anti-TNFα immunotherapy is only effective for <20% of patients. Mention potential resistance mechanisms or alternative pathways involved in inflammatory imbalance.

SB: We have added a few sentences there.

Line 171: Expand on "inflammatory mediators (TNFα, IL-6, IL-8)" by briefly describing their specific roles in the tumor microenvironment.

SB: We have discussed this in the original submission. I have added a reference as as a detailed discussion is beyond the scope of this review. New statement “What we know today is that inflammatory mediators (TNFa, IL-6, IL-8 and others) infiltrate the tumor microenvironment in an attempt to provide an attack against abnormal cells. However, what usually arises is genome instability and increased epigenetic activity driven by inflammation (https://www.nature.com/articles/s41392-021-00658-5).”

Line 175: Explain how tumor-associated macrophages (TAMs) and cancer-associated fibroblasts (CAFs) promote tumor growth with a specific example.

SB: I have explained more here.

Line 282-283: Explain how TAK1 links to both TLRs and NOD2 to drive inflammatory signals.

SB:  I have added a few more sentences here and more TAK1 biology is explained on the bottom of page 8 and 9.

Line 344: Briefly explain how TNFα, TRAFs, and TLR pathways differ in their contributions to NFκB modulation to provide context.

SB: This topic is discussed in detail in the literature and is beyond the scope of this review. A few sentences will not give justice to how varied these are connected to Each other.

Line 355: Explain why TLR7/TLR9 activation contributes to treatment challenges in ABC-DLBCL, with a focus on therapeutic implications.

SB: I have added the following to explain the role of TLR7/9 and provided a reference. “leading to accelerated growth (reviewed recently in 2023 by LeÅ›niak  et al., https://www.mdpi.com/2076-393X/11/2/27). TLR signaling have varied effects on biology with inflammation as a major effect of TLR activation. However, the strength of signal of this response can have tumor promoting effects in the tumor microenvironment and thus can affect the proliferative capacity of that cell. As such, it is a difficult disease to manage and treat.”

Line 364: Explain why Foxp3+ Treg cells are significant in preventing inflammation and obesity.

SB:  I have added “Foxp3+ CD3 cells are important in maintaining tolerance to commensal bacteria in the intestines and for preventing excessive inflammation in the gut.”

Line 383: Replace "hijack the energy generation machinery of mitochondria" with "reprogram mitochondrial energy production" to maintain a professional tone.

SB: We thank the reviewer for this change.

Line 471: Clarify the contradictory roles of iNOS by providing an example of its anti-cancer effects and discuss potential factors that dictate its dual functions.

SB: I have added lines to clarify this point.

Line 514: Briefly explain the JAK/STAT pathway and its role in inflammation and cancer to provide context.

SB: We have mentioned links to Jak/stat as with other pathways  and Jak/stat signaling is reviewed in detail in the literature. We feel that the addition of explaining Jak/stat may distract from the primary goal of this review.

Line 532: Discuss leptin’s potential role as a biomarker for low-grade inflammation in obesity and cancer.

SB: Page 15 has a whole section on leptin and explains the role of leptin in inflammation and cancer in great details.

Reviewer 2 Report

Comments and Suggestions for Authors

The manuscript introduces the concept of 'ImmunoMet Oncogenesis,' emphasizing the interplay between inflammation, metabolism, and proliferation as key drivers of cancer progression

However the narrative review is difficult to read:

  • I recommended to reorganize the content following the classical structure of a scientific review: Introduction, Methods, Results, and Discussion. Even if it is not a SR, methods are needed in order to explaining how the molecular pathways discussed were selected and analyzed, ensuring methodological transparency.
  • Focus on the novel aspects of the ImmunoMet Oncogenesis model and emphasize how it integrates these pathways into a unique conceptual framework.
  • Prioritize discussing less characterized molecular drivers and their roles in linking inflammation, metabolism, and proliferation.
  • Limit repetitive explanations of foundational cancer biology to avoid diluting the manuscript’s innovative contributions.
  • Some key references cited are outdated or lack the latest insights. For the purpose include and discuss the following recent publications: doi: 10.3390/cancers16223766. - doi: 10.3390/biom14030315. - doi: 10.3390/clinpract14020039. - doi: 10.1016/j.euros.2023.11.003 . The suggested covers different cancer topic but are all connected with your review proposal. 
  • Shorten sentences and make them more readable. 

minor:

  • The affiliations should be formatted

Author Response

Reviewer #2

We thank the reviewer for their comments and suggestions. We have made the required corrections as indicated below. 

  • I recommended to reorganize the content following the classical structure of a scientific review: Introduction, Methods, Results, and Discussion. Even if it is not a SR, methods are needed in order to explaining how the molecular pathways discussed were selected and analyzed, ensuring methodological transparency.

SB: We thank the reviewer for this comment. I have added some of these headers and new ones in section 4 to make it easier to understand those concluding concepts. I hope this is acceptable to the reviewer.

  • Focus on the novel aspects of the ImmunoMet Oncogenesis model and emphasize how it integrates these pathways into a unique conceptual framework.

SB: We have attempted to do this by focusing on each aspect of ImmunoMet Oncogenesis – inflammation, metabolism and proliferation. We have briefly described the prominent players but have highlighted emerging pathways that need to further explored. We have attempted to provide empirical evidence to link these emerging pathways to the concept of ImmunoMet Oncogenesis . We introduce the definition of ImmunoMet Oncogenesis  and end with concepts that could directly test the concept of ImmunoMet Oncogenesis . If we  remove content it will not be a strong review that has focused on the concepts of ImmunoMet Oncogenesis and given examples of ImmunoMet Oncogenesis drivers. I hope I have explained this well to the reviewer.

  • Prioritize discussing less characterized molecular drivers and their roles in linking inflammation, metabolism, and proliferation.

SB: We have written this review to think outside of the major players. We have introduced the major players but have focused on the “less characterized molecular drivers and their roles in linking inflammation, metabolism, and proliferation.” as stated by the reviewer.  We feel strongly that the ones we picked have arms towards inflammation, metabolism and proliferation and are worth exploring as biomarkers in cancer research

  • Limit repetitive explanations of foundational cancer biology to avoid diluting the manuscript’s innovative contributions.

SB: We were asked in a previous review at a different journal to bring in short discussions of the foundational aspects of cancer biology (such as the hallmarks of cancer). We have kept these discussion short with appropriate references (first 6 pages). From “Importance of NOD/RIPK2” onwards, we have focused on specific pathways that can contribute to ImmunoMet Oncogenesis . I hope this explains are choices to use foundational cancer biology concepts.

  • Some key references cited are outdated or lack the latest insights. For the purpose include and discuss the following recent publications: doi: 10.3390/cancers16223766. - doi: 10.3390/biom14030315. - doi: 10.3390/clinpract14020039. - doi: 10.1016/j.euros.2023.11.003 . The suggested covers different cancer topic but are all connected with your review proposal. 

SB: I have looked at all of these references suggested and have offer comments below:

doi: 10.3390/cancers16223766 – This review is focused on a global understanding of all cancers and drivers of malignancy. As such, we are not discussing specific cancers. Loss of Y is interesting but beyond the scope of this review as my review is quite long already.  I will consider it for a future specific report.

doi: 10.3390/biom14030315 - we are not discussing specific cancers. MUC1 is interesting but beyond the scope of this review as my review is quite long already.  I will consider it for a future specific report.

doi: 10.3390/clinpract14020039- we are not discussing specific cancers - tissue specific analysis is important for many diseases and very interesting but beyond the scope of this review as my review is quite long already.  I will consider it for a future specific report.

doi: 10.1016/j.euros.2023.11.003 – this article focuses on the microbiome and I have discussed this on pages 17 and 18.

  • Shorten sentences and make them more readable. 

SB: I thank the reviewer for this comment and have gone over the manuscript to fix run on or long sentences.

minor:

  • The affiliations should be formatted

SB: I have checked these and all appear good. The journal will fix any issues here once the manuscript is accepted.

Round 2

Reviewer 1 Report

Comments and Suggestions for Authors

The article has been improved. Congratulations!